# Safety and Quality of Canned Sardines after Opening: A Shelf-Stability Study

**DOI:** 10.3390/foods11070991

**Published:** 2022-03-29

**Authors:** Rebeca Cruz, Vânia Pereira, Teresa Pinho, Isabel M. P. L. V. O. Ferreira, Carla Novais, Susana Casal

**Affiliations:** 1LAQV/REQUIMTE, Laboratório de Bromatologia e Hidrologia, Faculdade de Farmácia, Universidade do Porto, Rua de Jorge Viterbo Ferreira 228, 4050-313 Porto, Portugal; vaniapatricia_22@hotmail.com (V.P.); teresapinho847@gmail.com (T.P.); isabel.ferreira@ff.up.pt (I.M.P.L.V.O.F.); sucasal@ff.up.pt (S.C.); 2UCIBIO/REQUIMTE, Laboratório de Microbiologia, Faculdade de Farmácia, Universidade do Porto, Rua de Jorge Viterbo Ferreira 228, 4050-313 Porto, Portugal; casilva@ff.up.pt

**Keywords:** shelf life, canned sardines, food safety, food quality, refrigeration

## Abstract

This study aimed to define the shelf life of canned sardines after opening to increase consumer awareness of their quality and safety and reduce food waste. For this purpose, canned sardines (*Sardina pilchardus*) packed with different sauces were opened and stored at 4 °C for 7 days. Microbiological, sensorial, physical and chemical stability was monitored daily by standard methodologies. Results show that the overall quality and safety are highly dependent on the sauce type. To preserve their full quality, sardines in brine and in vegetable oil should be consumed up to 1 day after opening, while sardines in tomato sauce were stable for up to 3 days, although none were considered nonedible up to the 7th day. Many parameters demonstrated statistical differences and correlations with storage, although they were not as decisive as sensory evaluation. This integrated approach should be adopted by the food industry and regulating authorities to provide information to consumers regarding the quality and safety of handled goods.

## 1. Introduction

The food industry has been making great efforts to offer products that meet the highest quality and safety standards to protect the health of consumers, as well as to provide foods that are easy to consume and that are available worldwide and all year long. Furthermore, health organizations are unanimous in advising the regular consumption of fish as a vital part of the human diet [1,2]. According to the World Health Organization, regular fish consumption (i.e., 1–2 servings per week) protects against coronary heart disease and ischaemic stroke [2]. Moreover, each serving should provide an equivalent of 200–500 mg of eicosapentaenoic (EPA) and docosahexaenoic acids (DHA) to achieve clinical significance [2]. In addition to these macronutrients, fish is also a source of high-quality protein, lipid-soluble vitamins, microelements and other fatty acids with recognized health benefits [3]. Therefore, appropriate manufacture, handling and storage conditions are required to preserve the quality of the fish until it reaches consumers.

Fish spoilage is influenced by several post-mortem factors including endogenous enzyme activity, non-enzymatic lipid oxidation and browning, microorganism development and storage conditions [3,4,5]. Therefore, several preservation techniques are usually applied to maintain the nutritional features of fish. Fish canning is a classical and widely accepted approach for long-term preservation [4]. The extensive heat treatment through the steps of cooking and sterilization should inactivate enzymes, microorganisms and their spores, creating a different product and extending fish shelf life. Although the freshness of fish should be ensured throughout the canning process, losses in quality and breakdowns of beneficial nutrients, with sensorial implications, inevitably occur during processing, and more slowly throughout storage. These effects are widely documented in the literature [6,7,8].

Depending on can size, a consumer might not use up its full content. This may lead to food waste or consumption of a product with potentially less quality or unknown safety, as no studies evaluating the shelf life of canned products after opening are available. Hence, this work aimed to evaluate the stability of different canned sardine products after opening with storage at 4 °C for up to seven consecutive days. To support this study, several microbiological, physical, chemical and sensorial parameters were used to evaluate the quality and safety of the samples under study. This experiment was established to define the limit of consumers’ acceptability for each product and, simultaneously, to help understand which of the parameters are determinant to estimate food spoilage.

## 2. Materials and Methods

### 2.1. Standards and Reagents

Biogenic amine standards (cadaverine dihydrochloride (CAD), histamine dihydrochloride (HIS), L-proline, putrescine dihydrochloride (PUT), serotonin (SER) spermidine trihydrochloride (SPD), spermine tetrahydrochloride (SPM) tyramine hydrochloride (TYR)), and internal standard (1,7-diaminoheptane) were acquired from Sigma-Aldrich, Madrid, Spain, as were bis-2-ethylhexylphosphate (BEHPA), boron trifluoride (14%), Tween 80^®^, 1,4-dioxane, triundecanoin (used as internal standard), and tocopherols standards. Acetonitrile, methanol and n-hexane (all LiChrosolv^®^, Darmstadt, Germany), ammonia, butylated hydroxytoluene (BHT), chloroform, hydrochloric acid, iodine, perchloric acid, phenolphthalein, phosphoric acid, silicone anti-foaming agent, sodium carbonate, 1,1,3,3-tetraethoxypropane (TEP), toluene, and trichloroacetic acid were purchased from Merck (Darmstadt, Germany). Anhydrous sodium sulphate, 1,4-dioxane, glacial acetic acid, potassium hydroxide, ethanol, sodium chloride and sodium hydroxide were supplied by VWR (Darmstadt, Germany). Other standards and reagents were acquired from several suppliers: as boric acid (Scharlab S. L., Madrid, Spain), cyclohexane (Roth, Germany), peptone (Frilabo, Maia, Portugal), 2-propanol (Roth, Karlsruhe, Germany), Supelco 37-FAME Mix (Supelco, Madrid, Spain), 2-thiobarbituric acid (Panreac, Barcelona, Spain), and tocol (Matreya, Pleasant Gap, PA, USA). Deionized water of 0.055 mS.cm^−1^ was obtained with a Seralpur Pro 90CN from Seral (Ransbach-Baumbach, Germany).

### 2.2. Sampling and Sample Preparation

Canned sardines (*Sardina pilchardus*), gutted and without head and tail fin, were provided by Ramirez & C^a^ (filhos) S.A. (Portugal) in easy-to-open metal containers (aluminium or tinplate) which had been hermetically sealed and properly sterilized by heat treatment. Samples comprised sardines in light brine (SB), sardines in vegetable oil (SO) and sardines in tomato sauce (ST). Appendix A shows the relative composition, net weight/drained weight and shelf life of products, as indicated by the manufacturer. Forty cans of each product type were tested. Three of them were used for the physicochemical assays and a further three for the sensorial analysis, over six sampling days (D0, D1, D2, D3, D4 and D7 = 3 × 2 × 6). Four further cans were combined and subsamples were used for the microbiological assays, following the experimental design summarized in Figure 1.

Drained sauce from one can of each type of fish preserve was also collected over six sampling days (D0, D1, D2, D3, D4 and D7 = 1 × 6).

#### 2.2.1. Physicochemical and Sensorial Analysis

At D0, all cans were opened, and their contents were individually transposed by inversion into a plastic container suitable for food storage. Except for D0 samples that were analysed immediately, all samples were kept refrigerated at 4 °C. Each day, samples were left to reach room temperature (for about 15 min) and either tasted by the panellists or homogenised in a food processor for the physicochemical tests, after draining the sauce, according to the usual patterns of canned fish consumption.

#### 2.2.2. Microbiological Analysis

Before opening, all cans were disinfected with a solution containing iodine (4%, *m/v*) and ethanol (70%, *v/v*) and manipulated under asepsis conditions, as recommended by Landry et al. [9]. For each product (SB, SO and ST), four cans from the same batch were combined in a sterile bag and portions of the composite samples (10 g) were placed in four separate sterile containers representing each of the different storage days (D0, D1, D2, and D7). Excepting D0 samples, which were analysed immediately, all the remaining samples were kept refrigerated at 4 °C until analysis.

### 2.3. Physicochemical Analysis

#### 2.3.1. Colour Measurement

Instrumental colour was measured on the surface of three preserved sardine fillets (3 measurements on the skin side and 3 measurements on the inner side) from three cans per day. Drained sauce from one can was subjected to one colour measurement per day. The analysis was conducted with a Minolta CR-400 colorimeter (Konica Minolta Optics Inc., Japan) with colour coordinates computed on the CIELAB scale. Colour results were expressed as tristimulus parameters: lightness (L*), redness (a*), and yellowness (b*). The total colour variation (ΔE) was also determined according to [10].

#### 2.3.2. Estimation of Fat, Fatty Acids, and Vitamin E

The homogenized sample (0.5 to 1.0 g), 2 glass beads, BHT solution (50 µL, 10 mg.mL^−1^ in methanol), triundecanoin (20 µL, 15 mg.mL^−1^ in hexane) and tocol (20 µL, 1 mg.mL^−1^ in hexane) were accurately transferred to a centrifuge tube. The extraction of lipid material was performed by cold extraction using isopropanol and cyclohexane, as described by Cruz et al. (2013) [11]. Fatty acids were analysed by gas chromatography with FID detection, after conversion into their fatty acid methyl esters. Vitamin E was analysed by normal-phase liquid chromatography with fluorescent detection. Fatty acid results are expressed in g.100 g^−1^ after proper calibration with standards. Total fat corresponds to the sum of total fatty acids with an LOQ of 10 mg.100 g^−1^ and a repeatability lower than 5%. Vitamin E was evaluated by the internal standard method based on the fluorescence data, and results are expressed on a fresh weight (FW) basis in milligrams per 100 g of sample, with a LOQ of 0.04 mg.100 g^−1^.

#### 2.3.3. Thiobarbituric Acid Reactive Species

The thiobarbituric acid reactive species (TBARS) test was based on the method developed by Papastergiadis, et al. [12], with minor modifications. Briefly, the sample (2.0 g) was homogenized with trichloroacetic acid (7.5%, *m*/*v*) and centrifuged (5000 rpm, 5 min). Then, the supernatant (500 μL) was mixed with an equivalent volume of 2-thiobarbituric acid solution (46 mM in glacial acetic acid) and incubated at 95 °C for 35 min in a water bath, with the absorbance recorded at 532 nm. Quantification was performed by external calibration with TEP solutions (0.5–15 µM) and the results were expressed on a FW basis as milligrams of malondialdehyde (MDA) per kilogram of sample.

#### 2.3.4. Total Volatile Basic Nitrogen

Total volatile basic nitrogen (TVB-N) was estimated based on the method defined in (EU, 2005) with slight adjustments. The sample (5.0 g) was homogenized with perchloric acid (90 mL, 6%, *m/v*) in an Ultra Turrax system (T25 Basic, Janke & Kunkel IKA, Germany), followed by filtration. In a steam distillation tube, phenolphthalein, silicone anti-foaming agent and sodium hydroxide (5 mL, 30%, *m*/*v*) were added to the filtrate (40 mL). The distillation outflow tube was collected in boric acid solution (3%, *m*/*v*) containing the Tashiro indicator solution and titrated with hydrochloric acid (0.01 M). The results were expressed as milligrams of N per 100 g of sample on a FW basis.

#### 2.3.5. Biogenic Amines

The internal standard (1,7-diaminoheptane, 60 µL, 2.5 mg.mL^−1^ in hydrochloric acid 0.1 M) was added to a 5.0 g portion of each sample. Then, extraction with trichloroacetic acid solution (5%, *m/v*) followed by ion-pair clean up with BEHPA was performed as described in a previous paper [13]. Dansyl chloride (1.0 mL, 10 mg.mL^−1^ in acetone) was added to 500 µL of the extract and the derivatization procedure was performed at 60 °C for 30 min. Liquid chromatography with fluorescent detection was used for the analysis (Jasco, Japan) and separation was achieved with a Gemini NX reverse phase column (150 × 4.6 mm, 5 μm, Phenomenex, Torrance, CA, USA). Quantification was based on the fluorescence signal (Ex = 254 nm, Em = 500 nm) response, using the internal standard method with the results being expressed on a FW basis, with a LOQ ranging from 0.02 to 0.05 mg.100 g^−1^. The biogenic amine index was estimated as the sum of PUT, CAD, TYR and HIS according to Prester (2011) [14].

#### 2.3.6. Other Measurements

The pH was measured at room temperature (23 ± 3 °C) with a pH electrode suitable for solid samples (pH meter GLP 21, Crison, Spain). The moisture content was evaluated with an infrared moisture analyser (Scaltec SMO 01, Heiligenstadt, Germany) at 105 °C.

### 2.4. Microbiological Analysis

Initial suspension and decimal dilutions for microbiological examination were performed with peptone water (100 mL, 0.1%, *m/v*) containing sodium chloride (0.85%, *m/v*) as described in ISO 6887-1:2017 [15]. The diluent was added to 10 g of the sample (1:10) and the mixture was homogenized in a Stomacher for 2 min. For high-fat samples, Tween 80^®^ (0.5%, *m/v*) was also added to the diluent as recommended in ISO 6887-1:2017 [15]. The total counts of aerobic mesophiles (Plate Count Agar, PCA; pour plate method; 30 °C for 5 days), aerobic psychrophiles (PCA; spread plate method; 15 °C for 10 days), aerobic thermophiles (PCA; spread plate method; 55 °C for 10 days) as well as anaerobic mesophilic sulphite-reducing bacteria (Iron Sulphite Agar, ISA; pour tubes method; 37 °C for 10 days), and anaerobic thermophilic sulphite-reducing bacteria (ISA; pour tubes method; 55 °C for 10 days) were investigated on the basis of official standards [16,17,18,19]. Results were expressed as colony-forming units (CFU) per gram of food [20]. Data were interpreted following the available criteria for ready-to-eat products from: a) the Health Protection Agency of the United Kingdom [21] (i.e., Category 1—“Ambient stable canned, bottled, cartonned and pouched foods immediately after removal from container” for D0 samples, and Category 5—“Cooked foods chilled but with some handling prior to sale or consumption” for D1, D2 and D7 samples) for the enumeration of aerobic microorganisms; b) the Portuguese National Institute of Health “Doutor Ricardo Jorge” (Group 1—i.e., meals, sandwiches, cakes, sweet desserts with fully cooked ingredients, or added dried herbs, dehydrated or treated with ionizing radiation, UHT products and industrialized mayonnaise) for the enumeration of sulphite-reducing anaerobic microorganisms [22].

### 2.5. Sensorial Analysis

Sensorial analysis was conducted at Ramirez & C^a^ (filhos) S.A. by a group of seven panellists trained for descriptive analysis according to the guidelines defined in ISO (2012) [23]. During training sessions, panellists proposed several attributes and those that were found to be redundant descriptive terms were removed. Concerning sardine sensory evaluation, the final attributes for visual appearance were loss of shine, fibrousness, as well as exterior and interior colour; for odour they were fermentation and putridness; for flavour they were acidity and putridness; and for texture it was fibrousness. Concerning sauce appearance, the attributes were turbidity and thickness. Ballot anchors were established for those attributes that were fitted on a structured scale (six points). All sensory evaluations were established in comparison with D0 (control), where 0—not different; 1—slightly different; 2—a little different; 3—different; 4—very different; 5—extremely different; 6—completely different. In each evaluation session, assessors received a list of attributes that varied according to the sample (i.e., fish or sauce) under study. All samples were presented to assessors in plastic food containers identified with a 3-digit code. Assessment was performed individually under white light at room temperature. Each assessor received filtered water to cleanse his or her palate between tastings.

### 2.6. Statistical Analysis

Dependent variables were studied using Kruskal–Wallis tests, and, when normal distribution of the residuals was not confirmed, by a Shapiro–Wilk test, followed by Mann–Whitney U tests if significant statistical differences were found. If normal distribution of the residuals was confirmed by Shapiro–Wilk test, dependent variables were studied using one-way ANOVA (with/without Welch’s correction) for independent samples, followed by means comparisons by Duncan’s or Dunnett’s tests, depending on whether or not homogeneity of variance was verified by Levene’s test, respectively. Pearson correlations were established between storage time and sensorial and instrumental data. Statistical analyses were performed at a 5% significance level, using SPSS software (version 26.0, IBM Corporation, New York, NY, USA).

## 3. Results and Discussion

### 3.1. Physicochemical Quality and Safety

#### 3.1.1. Nutritional Composition

Physicochemical alterations during the refrigerated storage of canned sardine products are detailed in Table 1. Moisture remained stable over the storage period at refrigeration temperatures for all canned products, with minor oscillations (no statistical significance) (Table 1). Nevertheless, moisture was higher in SB, followed by ST and SO. These differences can be related to the type of sauce (e.g., water, oil), the incorporation of vegetables (e.g., tomato), the compositional variation of fish (e.g., season, gender), and technological processing [24,25].

Concerning the evaluation of pH, small but significant differences were found in SB and SO and a weak negative correlation with storage time was observed for the former (Table 1). In most fish species, post-mortem pH is commonly between 6.0 and 6.8, based on the initial glycogen content (affected by the nutritional status and activity of fish before death) and accumulation of L-lactate [26]. Both SB and SO had an initial pH of 6.6–6.7, with minor variations with storage, but statistical differences between D0 and D1. ST had a lower pH due to the sauce (5.9), which was preserved up to D7. Therefore, our results are close to those reported in the literature and the variations observed cannot be regarded as an effective measure of spoilage [26].

In general, no significant differences were verified for total fat content, based on total fatty acid sum, during storage time for any of the canned products, except for a unique reduction in ST on D1 (Table 1), almost certainly resulting from fish-intrinsic features. However, by comparing the total fat of SB (4.1% to 6.1%) with that of ST (7.6% to 10.4%), we can see that tomato sauce highly contributes to the total fat content of the edible portion of canned products, being close to values for SO (11.8–13.8%), where lipid enrichment is expected. According to Selmi et al. (2008) [27], the canning process does not affect sardine lipid levels, although the fatty acid profile is changed by the absorption of coating oil and tomato sauce during sterilization. In our study, typical LC-PUFAs of fish (e.g., EPA, DHA and docosapentaenoic—DPA) were highly represented in all the samples, despite the clear influence of the fatty acids from the oil in SO, and from the tomato sauce in ST. On average, total PUFAs in the SO fat corresponded to 51%, with 20% of the sum of DHA, EPA, and DPA, alongside with 26% of linoleic acid, 17% of oleic acid and 3% of α-linolenic acid (data not shown). On the other hand, in ST, LC-PUFAs represented 35% of the total fatty acids, close to the 38% observed in SB, the latter without influence from external lipids. However, in absolute amounts, the sauce has a highly relevant role in LC-PUFAs preservation, with higher amounts in ST at D0 (3.62 g.100 g^−1^), followed by SO (2.76 g.100 g^−1^) and SB (1.93 g.100 g^−1^), indicative of protective effect in ST and higher degradation in SB. Still, the preservation of LC-PUFAs content in all canned products studied during storage indicates that the integrity of fish fillets protects the flesh from oxidation during storage, while the sauce has a stronger impact before opening.

No statistical differences were observed in the vitamin E content of canned products during storage (Table 1) but the differences between the products were again highly relevant. Of all samples, SO showed the highest vitamin E content (4.9 mg.100 g^−1^), with ɣ-tocopherol accounting for 60% of the total, derived from the vegetable oil used (soybean oil), while in ST, α-tocopherol represented more than 94% of total vitamin E (data not shown), with a total content of 1.61 mg.100 g^−1^, still near to fresh sardines (2.1 mg.100 g^−1^) [28,29]. However, the amounts of vitamin E in SB were reduced (0.23 mg.100 g^−1^), in line with the reduction in LC-PUFA previously discussed. Considering the antioxidative properties of vitamin E, the reduced content of tocopherols might be an indicator of the greater oxidative impact of the salted brine used in canning.

#### 3.1.2. Lipid Oxidation and Proteolysis

The thiobarbituric acid-reactive species index is considered as a standard marker for lipid peroxidation-induced oxidative stress [30]. By-products of lipid oxidation in canned sardines are shown in Table 1. No statistical differences were found in SB and SO, which presented average TBARS values ranging from 3.26 to 4.40 mg.kg^−1^ and 3.60 to 4.22 mg.kg^−1^, respectively. As confirmed by its lower vitamin E content, the brine from SB contains sodium chloride that may favour the oxidation of highly unsaturated fatty acids, because of the metal ions released by haemoprotein complexes, but also due to the pressure generated at the sterilization step that can increase TBARS content [31]. Initial TBARS amounts in ST were considerably lower than those of SB and SO, with a significant correlation with storage from D3. Despite its higher LC-PUFAs content and lower amounts of vitamin E, ST still showed the lowest degree of lipid oxidation. This evidence points to a likely additional antioxidant potential coming from the tomato sauce, probably from lycopene, which reduces the lipid oxidation of ST fillets. A TBARS value of 1.0 mg.kg^−1^ is commonly considered as the threshold of lipid oxidation [30]. Therefore, except ST from D0 to D2, all other samples tested revealed significant lipid oxidation levels (Table 1).

TVB-N is an internationally recognized indicator of food spoilage derived from protein degradation and is determined by the measurement of trimethylamine (resulting from the microbial deterioration of trimethylamine oxide), dimethylamine (produced by autolytic enzymes during storage), ammonia (produced by deamination of amino acids and nucleotides) and other low molecular weight amines generated by the decarboxylation of amino acids [26]. The estimation of TVB-N in canned sardines is detailed in Table 1. In contrast to lipid oxidation, no statistical differences were verified for SO and ST, while in SB significant differences were observed from D1 (Table 1). Such differences in SB were also accompanied by a significant positive correlation with storage time.

Commission Regulation (EC) No. 1022/2008 sets a TVB-N limit of 35 mg.100 g^−1^ of flesh for some species of fresh fish [32]. However, there are no legal limits of TVB-N for processed fish. Nevertheless, at D7, protein degradation in SB was very close to the maximum values recommended by Aubourg (2001) [4] for preserved fish (45.0 mg.100 g^−1^ of fish). As confirmed by Shakila et al. (2005) [33], after 6 h at room temperature, canned sardines (*Sardinella gibbosa*) showed a TVB-N content ranging from 20 to 30 mg.100 g^−1^. Another study developed by Losada et al. (2006) [7] showed the differences between sardines (*Sardina pilchardus*) kept in ice as a preliminary treatment for the canning process and verified that TVB-N ranged from 52.7 to 55.7 mg.100 g^−1^ after 5 days of storage. Hence, our figures are within those found in the literature, although storage conditions (e.g., time and temperature), technological process, and fish species, among other factors, differ between studies. As far as ST is concerned, no significant differences were observed with storage time and the lowest TVB-N content of all canned products was recorded (Table 1), thus remaining below the recommended limits [4].

#### 3.1.3. Biogenic Amines

Biogenic amines are organic bases with biological activity, which are mainly produced because of amino acid decarboxylation [34]. They also play an important role as a marker of microbiological contamination. Thus, its monitoring is vital when estimating the shelf life of foodstuffs. The formation of biogenic amines in canned sardines throughout refrigerated storage is shown in Figure 2. In SB, SPD was present in greater amounts at all storage days, followed by PUT, SPM and HIS, with significant differences from D1. Cadaverine was not produced in SB products. Moreover, significant and strong positive correlations with storage time were obtained for PUT (r = 0.932, *p* < 0.001), HIS (r = 0.745, *p* < 0.001), SPD (r = 0.745, *p* < 0.001), total amines (r = 0.878, *p* < 0.001) and the biogenic amine index (r = 0.936, *p* < 0.001).

In general, SO showed a higher content of individual and total biogenic amines in comparison to SB (Figure 2). Spermidine was the most abundant, followed by SPM, PUT, HIS and CAD, with statistical differences observed at D1, as for SB samples. Furthermore, significant and positive correlations with storage time were obtained for CAD (r = 0.472, *p* < 0.05), HIS (r = 0.548, *p* < 0.05), total amines (r = 0.532, *p* < 0.05) and the biogenic amine index (r = 0.708, *p* < 0.001). In [28], a significant increase in HIS (from 1.6 to 2.3 mg.kg^−1^) during storage of SO samples was also reported.

In ST samples, PUT, CAD, HIS, SPD and SPM were identified, and statistical differences were confirmed from D1 (Figure 2). Significant and strong positive correlations with storage time were obtained for PUT (r = 0.727, *p* < 0.051), CAD (r = 0.780, *p* < 0.001), HIS (r = 0.789, *p* < 0.001), total amines (r = 0.710, *p* < 0.001) and the biogenic amine index (r = 0.896, *p* < 0.001). Despite the slightly lower total amine content in SO, the biogenic amine index was considerably higher in ST due to the greater production of PUT (Figure 2). Ali et al. (2011) [35] showed high amounts of PUT in fresh and processed tomatoes reaching up to 41.1 mg.kg^−1^. Therefore, the lower quality of ST is likely more related to the presence of tomato sauce rather than the quality of the fish.

According to Commission Regulation 1441/2007, the maximum permitted limit for HIS in “Fishery products from fish species associated with a high amount of histidine” (e.g., Scombridae and Clupeidae) lies between 100 and 200 mg.kg^−1^ [36]. Further, the US Food and Drug Administration reported levels of HIS of 50 mg.kg^−1^ and 500 mg.kg^−1^ as indicators of decomposition and toxicity, respectively [34]. The results from the current study for all canned sardines are far below the recommended limits mentioned above, and thus, a 7-day storage time can be considered as safe regarding biogenic amine production.

### 3.2. Instrumental Colour

Despite instrumental measurement alone not being capable of indicating consumer acceptance or rejection [37], it is a useful tool to corroborate the data obtained from the sensorial evaluation. The results of instrumental colour for drained sauce and preserved sardine fillets are presented in Table 2 and Table 3, respectively. As regards SB, significant differences during refrigerated storage of canned sardines were confirmed for L* (external and internal sides) and coordinates a* and b* (internal side). At D0, lightness was higher in the external side due to the skin and scales that are typical features of this type of product. On the contrary, at D0 the internal side presented lighter shades, less lightness, and red notes near the spine. A gradual decrease in all colour coordinates was observed in the internal side until D7, although the reduction in redness was more noticeable in comparison to yellowness, as confirmed by a lower and more statistically significant Pearson correlation (Table 1). These findings are coherent with the scores from the sensorial analysis (see Section 3.4, Sensorial performance), but also with colour modifications due to oxidation and myoglobin changes reported in the literature [38].

Regarding the drained sauce from SB, differences in instrumental colour were statistically significant only for the parameters a* and b*, which showed positive and negative correlations with storage time, respectively (Table 2). Colour variation (ΔE) was detected at D1 and continued to increase with storage time, reaching its maximum at D4, likely related to the presence of small particles of fish skin in the brine. According to Pathare et al. (2013) [10], differences in perceivable colour can be analytically classified as very distinct (ΔE > 3), distinct (1.5 < ΔE < 3) and small difference (1.5 < ΔE). Therefore, in our study, the colour of SB fillets was very distinctive from D1, while the sauce revealed small differences during storage, except for D4.

As for SO products, lightness was higher on the external side in comparison to the inside, as expected, but always lower than the lightness of SB (Table 1). In addition, a significant negative Pearson correlation was verified for external L*, which is coherent with the loss of shine verified in sensorial evaluation (Figure 3). Yellowness decreased significantly at D1 on the internal side, whereas only at D4 a reduction was observed externally (Table 1). Regarding redness, significant differences were confirmed on the internal side, likely owing to the irregular morphology (e.g., reddish areas and dark spots) near the spinal cord. Concerning the drained sauce from SO, instrumental measurement showed significant differences only for the coordinate a*, in which the values presented a negative and highly significant correlation with storage time (Table 2). This behaviour may be related to the increase in turbidity, for which higher figures were registered at D4 and D7 by both instrumental and sensorial analysis. In general, the total colour difference was small in the sauce, but very distinctive in the fish (Table 2 and Table 3).

ST also showed higher lightness on the external side, but it was lower than that measured in SB and SO (Table 1). According to Qiao et al. (2002) [39] and Kilinc et al. (2008) [6], L* values should be included in three different ranges: (i) lighter-than-normal or light (L* > 53); (ii) normal (48 < L* < 51), and (iii) darker-than-normal or dark (L* < 46). In our study, sardine fillets showed lightness oscillating during the storage period tested, although it remained in the category light (L* > 53), which is coherent with the findings of Kilinc et al. (2008) [6]. Statistical differences in the coordinates a* and b* were more evident for the inner part of the fillet, and no distinction was verified for ST (Table 3 and Table 4). ST fillets presented high ΔE values on the external side due to greater exposure to oxidant agents in comparison to the internal side (Table 1). Overall, all samples show significant differences in ΔE values at D1 in comparison to D0 for the interior portion of fillets. This likely indicates that after one day of opening, canned sardines show a quality deterioration, as confirmed by chemical analysis.

### 3.3. Microbiological Quality

The results for microbiological stability under refrigeration are detailed in Table 4. At D0, the enumeration of colonies was below the detection limit of the method (<10 CFU.g^−1^ of aerobic mesophiles or sulphite-reducing anaerobic microorganisms; <10^2^ CFU.g^−1^ of aerobic psychrophiles or thermophiles). Based on these outcomes, the microbiological stability was classified as “satisfactory” according to established criteria for ready-to-eat foods previously described. Despite the satisfactory microbiological performance at D7 for all samples, aerobic psychrophilic and thermophilic microorganisms were present in sardine samples in light brine or vegetable oil (<4 × 10^2^ CFU.g^−1^), but not in tomato sauce. This shows that microorganisms are still able to grow during storage, thus reaching concentrations that are detectable by analytical methods. Considering the aseptic handling conditions during sampling, these bacteria may arise from the raw material (e.g., sporulated bacteria more tolerant to heat treatment). Oranusi et al. [40] investigated the profile of deteriorating psychrophilic, mesophilic and thermophilic microorganisms in different canned foods within the expiry date. At the time of opening, sardine samples had low microbial loads <10^2^ CFU.g^−1^, with the authors considering them as being within acceptable microbiological quality [40]. Oyelese & Opatokun (2007) monitored the microbial count and shelf life of canned sardines with 4 years of expiration date under room temperature and refrigerated storage conditions over 12 weeks [8]. No cultivable viable bacteria count was verified for cold-stored samples throughout the experiment, but in those at room temperature, a total count of 2.1 × 10^4^ CFU.g^−1^ was observed [8].

These results should be interpreted with caution because the microbial quality and safety of canned foods rely on numerous factors that include the quality of the raw material, heat treatment efficacy, storage temperature, and adequate food manipulation practices. Therefore, the results of the current study may not entirely reflect the reality of the domestic environment considering the aseptic measures employed herein to avoid cross-contamination and bias.

### 3.4. Sensorial Performance

The sum of attribute scores of the sensorial analysis of canned sardines are shown in Figure 3 based on sensory modifications throughout post-opening time. All samples showed acceptable sensorial attributes (score = 0) at the initial day of opening (D0, control) but a sensorial quality decay was observed with storage, likely due to progressing oxidation.

The overall appearance of food generates expectations and is normally the first feature that can be perceived by the senses of consumers. Besides colour, other appearance features include shape, transparency or observable textural properties. However, colour is the paramount attribute influencing the selection of foods and supplies the first information about product quality and safety. As far as fresh fish is concerned, its colour often changes from pink to white, though it can reach darker tones mainly due to oxidation, myoglobin changes and Maillard reactions [38]. However, a loss of brightness (appearance) was only detected by the panel at D4, likely due to the gloss granted by the oil in SO samples.

Odour plays a key role in consumers’ lives owing to its emotional and social aspects, stimulating or reducing appetite, but off-aromas and off-odours may also serve as good tools for consumers to identify spoiled foods. Fresh sardines are almost odourless due to the low quantity of volatiles, which mostly comprise trimethylamine and carbonyl compounds [41]. All samples showed low odour scores after opening, which are in accordance with our TVN-B observations, followed by small increments after D2. Regarding SO, small changes in odour and flavour from D2 forward were not perceived by all the assessors. This discrepancy increased the dispersion of the results and decreased the statistical significance of the data. On the other hand, SB exhibited a slow decay from D2 forward that can be explained by the fact that the skin protects the fillet from oxidation, which reduces the development of off-odours and off-flavours. ST showed significant changes only after D4, while deviations in some attributes like colour, putrid odour and putrid flavour were not detected even at D7. The tomato sauce gave a more intense flavour and odour, which helped to preserve fish deterioration, thus hindering sensorial perception.

Texture is also a vital quality characteristic of fish and other muscle tissues. Fresh fish is generally more tender than meat, since it contains less connective tissue than the meat of mammals and a lower degree of collagen cross-links [38]. Thermal processing, like canning, also has great impact on the tenderness of fish meat after processing [42]. Moreover, the free fatty acid content usually increases during fish storage, thus leading to protein denaturation, which, consequently, affects the texture and the water-holding capacity [38]. In the present study, low textural scores were obtained for all samples, although considerable differences were verified from D2 forward for SB and SO, but only from D4 forward for ST.

Considering the aforementioned observations, it is clear that sensory analysis should not focus exclusively on the deterioration of the product, but also on the perception by consumers, particularly through flavour and odour. Moreover, sensorial defects were identified at the same time, or sooner, than physicochemical modifications, including those that do not directly and necessarily compromise the safety of consuming these food products (e.g., ΔE or pH). Nevertheless, sensorial analysis provides quality discrimination based on a holistic approach of human senses, thus likely leading to an early rejection of foods and showing the relevance of its use in food quality and safety assessments.

## 4. Conclusions

Canned fish has a high shelf life, but after opening, it undergoes microbiological and chemical changes that compromise its stability. In this study, the shelf life of canned sardines after opening was strongly affected by the type of sauce and by storage time. Sardines canned in tomato sauce revealed the lowest lipid oxidation and can be refrigerated for up to three days without substantial chemical, microbiological or sensory modifications, while sardines in brine or vegetable oil should only be kept up to one day at 4 °C due to proteolysis and sensorial defects, respectively.

Sensorial, physical, and chemical analyses allowed the monitoring of food modifications with storage, although most parameters, including moisture, pH or even fatty acid composition, were not mandatory for the investigation of food shelf life. On the contrary, the sensorial analysis was the most relevant procedure for early assessment of shelf life. Considering that sensorial perception is the only mechanism that consumers possess to evaluate the quality of foods in the domestic environment, these results are of particular relevance. Hence, an integrated approach including not only microbiological and physicochemical studies, but also sensorial analysis should be adopted to ensure food safety. Overall, this study exposes how limited shelf life is after opening and provides relevant insights for the potential inclusion of additional additives that can extend shelf life after opening. Moreover, it enables producers to inform consumers about the estimated shelf life of opened products by including this information on the label. This will allow them to plan the use of food before it loses its quality and consequently reduce domestic food waste.

## Figures and Tables

**Figure 1 foods-11-00991-f001:**
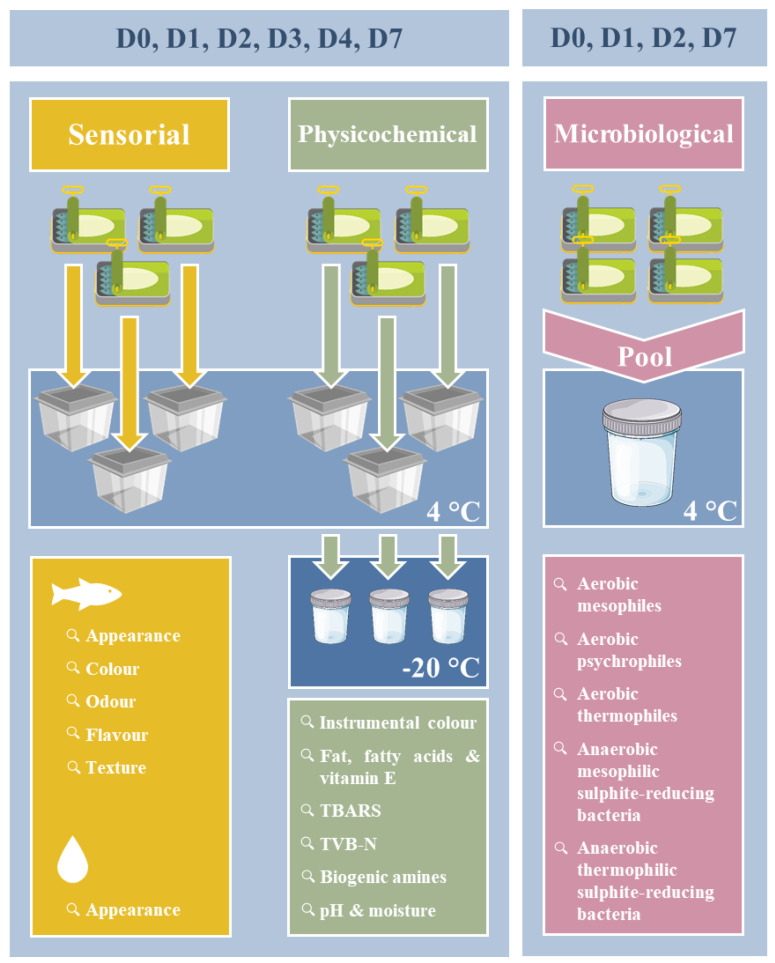
Experimental design of the evaluation of the overall quality of each canned sardine product, from the same batch, during refrigerated storage (D, day).

**Figure 2 foods-11-00991-f002:**
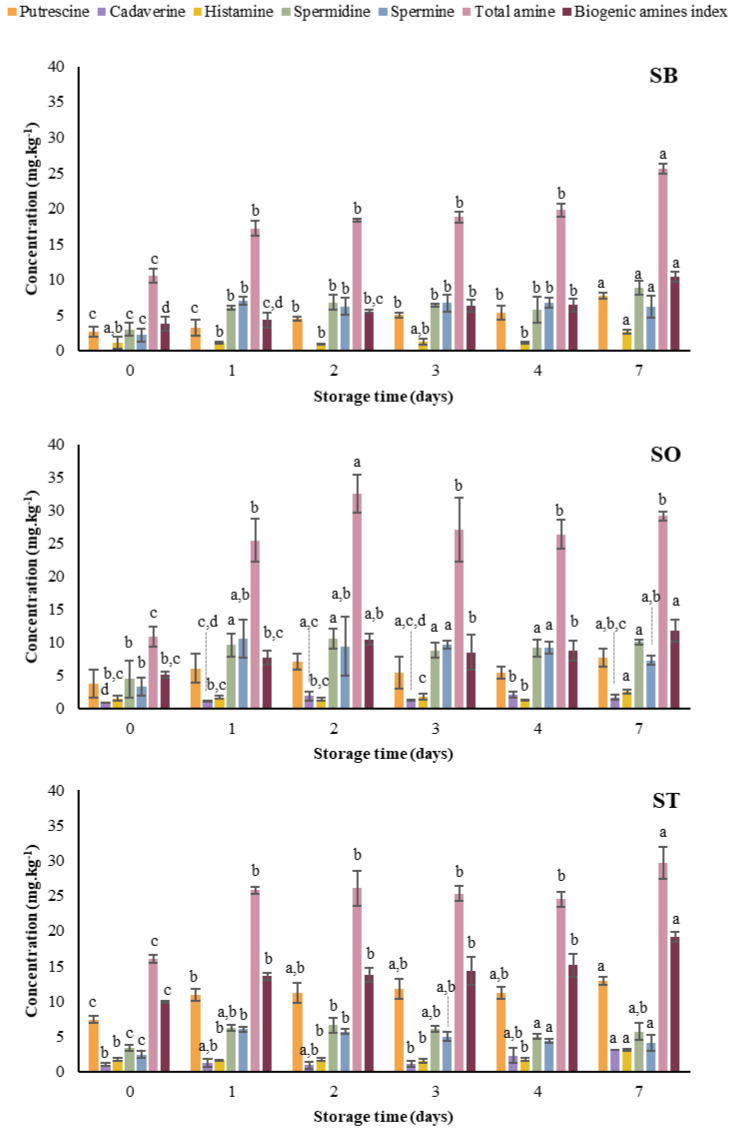
Biogenic amines, total amine, and biogenic amine index of canned sardines during storage at 4 °C. For each parameter, different letters between columns show statistical differences between means (*p* < 0.05).

**Figure 3 foods-11-00991-f003:**
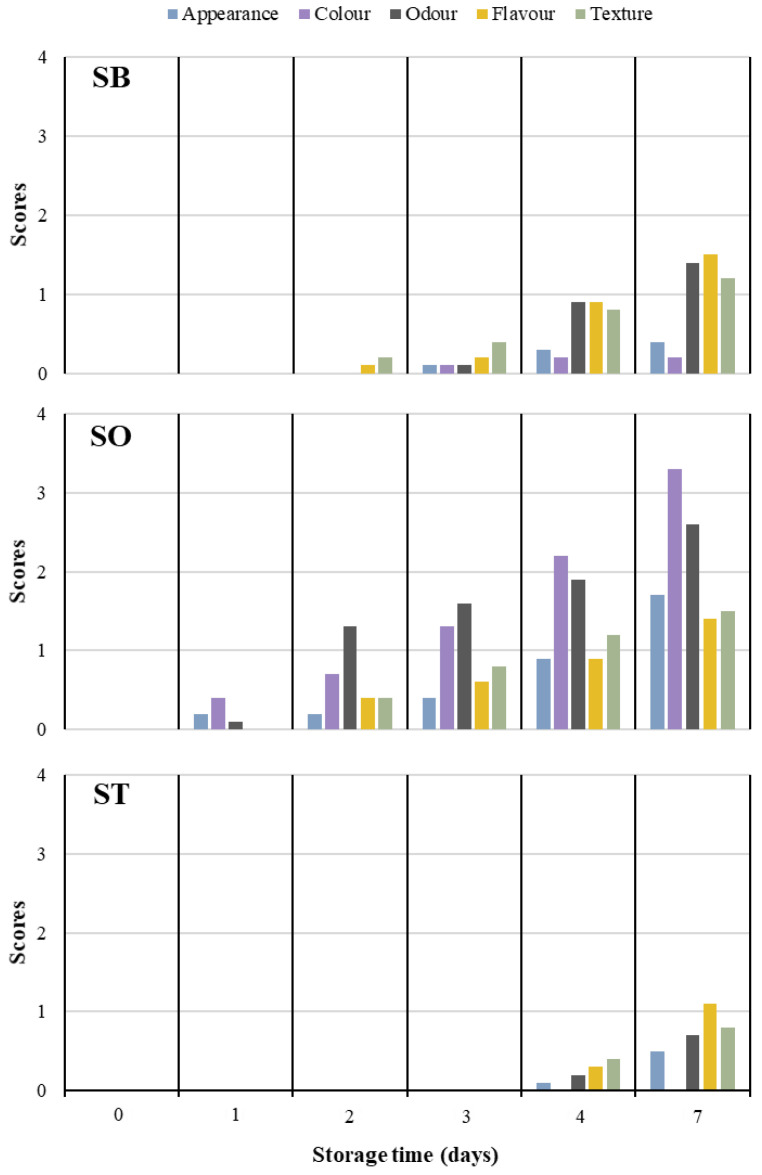
Sensorial analysis of canned sardines in light brine (SB), in vegetable oil (SO) and in tomato sauce (ST) during storage at 4 °C.

**Table 1 foods-11-00991-t001:** Physicochemical characterization of canned sardines during storage at 4 °C (mean ± standard deviation).

Parameters	Sample	Storage Time (Days)						*p*	r
		0	1	2	3	4	7		
Water (%)	SB	68.6 ± 1.0	69.9 ± 1.1	69.7 ± 1.5	67.6 ± 0.5	68.7 ± 1.2	70.1 ± 1.0	0.104	n.s.
	SO	58.8 ± 0.7	57.7 ± 2.5	61.7 ± 1.1	58.2 ± 2.2	58.2 ± 2.1	58.6 ± 0.2	0.128	n.s.
	ST	61.7 ± 1.1	62.9 ± 0.6	61.8 ± 0.4	62.5 ± 0.2	60.0 ± 0.7	62.0 ± 0.8	0.336	n.s.
pH	SB	6.65 ± 0.04 c	6.93 ± 0.04 a	6.97 ± 0.05 a	6.81 ± 0.05 b	6.76 ± 0.06 b	6.62 ± 0.06 c	˂0.001	−0.390 *
	SO	6.64 ± 0.04 a	6.56 ± 0.04 b	6.51 ± 0.04 b	6.45 ± 0.04 c	6.55 ± 0.06 b	6.52 ± 0.06 b	˂0.001	n.s.
	ST	5.95 ± 0.05	5.93 ± 0.07	5.89 ± 0.03	5.87± 0.11	5.94 ± 0.11	5.90 ± 0.08	0.484	n.s.
Fat (g.100 g^−1^)	SB	5.77 ± 2.90	4.08 ± 1.95	6.13 ± 0.65	5.52 ± 0.67	5.92 ± 1.21	4.30 ± 0.96	0.517	n.s.
	SO	13.68 ± 0.84	13.40 ± 1.35	12.63 ± 2.35	13.75 ± 1.00	11.81 ± 0.41	12.32 ± 1.38	0.448	n.s.
	ST	10.31 ± 1.28 a	7.62 ± 0.97 b	9.69 ± 0.81 a	9.69 ± 0.16 a	10.23 ± 0.65 a	10.36 ± 0.71 a	0.014	n.s.
LC-PUFAs (g.100 g^−1^)	SB	1.93 ± 1.07	1.48 ± 0.61	2.19 ± 0.24	2.01 ± 0.13	2.22 ± 0.40	1.71± 0.38	0.584	n.s.
	SO	2.67 ± 0.26	2.04 ± 0.49	2.14 ± 0.74	2.08 ± 0.39	2.06 ± 0.13	2.19 ±0.15	0.468	n.s.
	ST	3.62 ± 0.51 a	2.53 ± 0.52 b	3.49 ± 0.44 a	3.18 ± 0.24 a,b	3.69 ± 0.29 a	3.73 ± 0.20 a	0.020	n.s.
Vitamin E (mg.100 g^−1^)	SB	0.23 ± 0.12	0.20 ± 0.11	0.45 ± 0.06	0.33 ± 0.11	0.34 ± 0.13	0.22 ± 0.02	0.057	n.s.
	SO	4.90 ± 0.29	6.79 ± 2.09	4.63 ± 1.14	6.30 ± 1.27	5.36 ± 1.49	4.36 ± 0.55	0.208	n.s.
	ST	1.61 ± 0.11	1.68 ± 0.21	1.62 ± 0.14	1.46 ± 0.18	1.67 ± 0.23	1.25 ± 0.12	0.063	n.s.
TBARS (mg MDA.kg^−1^)	SB	3.54 ± 0.18	3.38 ± 0.82	3.26 ± 0.32	3.37 ± 0.14	4.24 ± 0.95	4.40 ± 0.88	0.203	n.s.
	SO	3.75 ± 0.45	4.04 ± 0.39	3.60 ± 0.32	3.91 ± 0.27	4.22 ± 0.71	3.99 ± 0.47	0.635	n.s.
	ST	1.96 ± 0.40 c	1.86 ± 0.23 c	2.50 ± 0.65 b,c	3.44 ± 1.29 a,b	3.35 ± 0.29 a,b	3.80 ± 0.37 a	0.012	0.742 **
TVB-N (mg N.100 g^−1^)	SB	40.2 ± 0.3 d	41.1 ± 0.3 c	41.0 ± 0.4 c	43.1 ± 0.3 b	38.9 ± 0.3 e	44.7 ± 0.3 a	˂0.001	0.586 *
	SO	43.8 ± 0.6	44.0 ± 1.1	41.9 ± 0.6	43.1 ± 1.2	44.3 ± 0.7	43.8 ± 0.8	0.053	n.s.
	ST	36.2 ± 0.4	36.0 ± 1.2	36.3 ± 0.6	35.6 ± 1.3	35.9 ± 0.4	35.3 ± 1.1	0.767	n.s.

Different letters in the same row show statistical differences between means (*p* < 0.05); Pearson correlation (r) is significant at the 0.01 (** *p*) and 0.05 (* *p*) levels. LC-PUFAs, long-chain polyunsaturated fatty acids; n.s., not significant; SB, sardines in light brine; SO, sardines in vegetable oil; ST, sardines in tomato sauce; TBARS, thiobarbituric acid reactive species; TVB-N, total volatile basic nitrogen.

**Table 2 foods-11-00991-t002:** Instrumental evaluation of the sauce of canned sardines’ storage at 4 °C (mean ± standard deviation).

Parameters	Sample	Storage Time (Days)						*p*	r
		0	1	2	3	4	7		
L*	SB	37.0 ± 1.6	36.0 ± 1.4	36.7 ± 2.8	36.3 ± 2.6	35.4 ± 1.5	35.9 ± 1.6	0.930	n.s.
	SO	30.1 ± 2.2	29.6 ± 1.5	31.8 ± 1.3	30.1 ± 1.7	31.4 ± 1.2	31.0 ± 1.1	0.475	n.s.
	ST	35.4 ± 1.3	35.0 ± 1.7	35.9 ± 1.2	36.3 ± 1.0	35.6 ± 1.2	35.2 ± 1.2	0.807	n.s.
a*	SB	−0.9 ± 0.1 b	−0.8 ± 0.1 a,b	−0.9 ± 0.0 b	−0.9 ± 0.0 b	−0.9 ± 0.0 b	−0.8 ± 0.1 a	0.021	0.470 *
	SO	−0.5 ± 0.1 b	−0.6 ± 0.1 a,b	−0.6 ± 0.1 a,b	−0.6 ± 0.0 b	−0.7 ± 0.2 a,b	−0.8 ± 0.1 a	0.029	−0.700 **
	ST	17.6 ± 1.1	17.7 ± 0.4	17.6 ± 0.9	17.2 ± 0.8	17.9 ± 1.0	17.5 ± 0.7	0.949	n.s.
b*	SB	4.3 ± 0.4 a	4.4 ± 0.3 a,b	4.2 ± 0.1 a,b,c	4.7 ± 0.5 a	3.7 ± 0.2 c	4.1 ± 0.0 b,c	0.029	−0.310 *
	SO	1.9 ± 0.1	1.8 ± 0.1	1.9 ± 0.2	1.8 ± 0.1	1.8 ± 0.1	1.9 ± 0.1	0.266	n.s.
	ST	19.6 ± 1.2	20.3 ± 2.0	20.4 ± 1.2	21.1 ± 0.3	20.9 ± 1.2	20.4 ± 0.8	0.746	n.s.
ΔE	SB	0.0 ± 0.0 c	1.1 ± 0.2 b	1.2 ± 0.5 a,b	1.5 ± 0.4 a,b	2.8 ± 1.2 a,b	1.3 ± 1.5 a,b	0.034	n.s.
	SO	0.0 ± 0.0	0.9 ± 0.3	1.8 ± 1.3	0.6 ± 0.4	1.4 ± 1.7	1.1 ± 0.8	0.339	n.s.
	ST	0.0 ± 0.0	2.3 ± 0.3	2.3 ± 1.1	3.0 ± 0.7	3.0 ± 1.2	2.3 ± 0.8	0.339	n.s.

Different letters in the same row show statistical differences between means (*p* < 0.05); Pearson correlation (r) is significant at the 0.01 (** *p*) and 0.05 (* *p*) levels. n.s., not significant; SB, sardines in light brine; SO, sardines in vegetable oil; ST, sardines in tomato sauce.

**Table 3 foods-11-00991-t003:** Instrumental colour of sardines during storage at 4 °C (mean ± standard deviation).

Parameters		Sample	Storage Time (Days)						*p*	r
			0	1	2	3	4	7		
Exterior	L*	SB	76.3 ± 5.1 a	72.4 ± 4.5 a,b	72.4 ± 4.3 a,b	73.4 ± 2.0 a	70.2 ± 4.4 b	69.9 ± 5.0 b	˂0.001	−0.369 **
		SO	73.1 ± 4.5 a	71.0 ± 8.8 a,b	76.4 ± 5.2 b,c	63.6 ± 4.3 c,d	61.9 ± 6.6 c,d	61.8 ± 5.7 d	˂0.001	−0.524 **
		ST	62.2 ± 5.9 a	56.8 ± 2.2 b	55.7 ± 6.6 b	57.9 ± 6.5 a,b	58.9 ± 5.9 a,b	58.4 ± 6.3 a,b	0.001	n.s.
	a*	SB	−0.8 ± 0.6	−0.7 ± 0.5	−0.8 ± 0.4	−0.8 ± 0.5	−0.7 ± 0.6	−0.7 ± 0.6	0.946	n.s.
		SO	−0.2 ± 0.7	−0.6 ± 0.9	−0.5 ± 0.9	−0.6 ± 0.7	−0.6 ± 0.7	−0.7 ± 0.9	0.393	n.s.
		ST	11.8 ± 3.7	10.9 ± 2.5	10.6 ± 3.7	10.2 ± 3.3	10.9 ± 3.6	10.3 ± 2.6	0.516	n.s.
	b*	SB	12.5 ± 2.9	13.9 ± 2.7	12.3 ± 2.1	12.1 ± 2.9	13.3 ± 3.0	12.1 ±2.9	0.130	n.s.
		SO	10.3 ± 2.9 a	9.9 ± 3.7 a	10.3 ± 2.5 a	10.1 ± 2.8 a	8.3 ± 2.4 b	8.0 ±3.0 b	0.006	−0.277 **
		ST	36.9 ± 6.4 a	32.9 ± 5.5 b	32.6 ± 6.5 b	32.7 ± 5.2 b	32.3 ± 5.9 b	33.0 ± 4.1 b	0.032	n.s.
	ΔE	SB	0.0 ± 0.0 b	8.7 ± 4.7 a	7.7 ± 4.3 a	6.9 ± 3.3 a	8.1 ± 5.0 a	8.0 ± 4.6 a	˂0.001	0.335 **
		SO	0.0 ± 0.0 b	9.5 ± 5.1 a	8.8 ± 5.3 a	10.7 ± 4.7 a	11.5 ± 5.3 a	13.0 ± 6.6 a	˂0.001	0.510 **
		ST	0.0 ± 0.0 b	11.4 ± 4.7 a	13.3 ± 7.6 a	12.3 ± 5.6 a	12.8 ± 6.8 a	12.3 ± 4.4 a	˂0.001	0.375 **
Interior	L*	SB	52.4 ± 3.9 a	46.7 ± 8.4 b	45.8 ± 6.0 b	46.6 ± 2.1 b	45.5 ± 4.2 b	46.6 ± 4.5 b	˂0.001	−0.325 **
		SO	48.9 ± 5.0 a	39.0 ± 6.0 b	43.0 ± 6.4 c	45.6 ± 5.4 c	44.5 ± 6.7 c	42.1 ± 6.4 b,c	˂0.001	n.s.
		ST	53.9 ± 3.6 a	48.0 ± 3.0 b	48.4 ± 5.4 b	48.1 ± 3.4 b	49.1 ± 1.8 b	48.9 ± 3.3 b	˂0.001	−0.216 **
	a*	SB	7.4 ± 2.6 a	6.4 ± 2.7 a,b,c	6.6 ± 2.0 a,b	6.9 ± 2.7 a,b	5.2 ± 1.8 b,c	5.0 ± 1.7 c	˂0.001	−0.314 **
		SO	6.9 ± 2.0 b,c	7.3 ± 2.2 a,b	8.1 ± 2.2 a	6.7 ± 2.0 b,c	5.9 ± 2.0 c	6.8 ± 1.9 b,c	0.006	n.s.
		ST	11.1 ± 2.2 a	8.3 ± 2.1 b	8.6 ± 2.0 b,c	9.4 ± 2.5 b,c	9.3 ± 2.0 b,c	9.8 ± 2.7 c	˂0.001	n.s.
	b*	SB	18.2 ± 2.0 a	16.0 ± 2.5 b	17.3 ± 2.8 a,b	16.1 ± 1.8 b	16.0 ± 1.4 b	16.7 ± 1.8 b	˂0.001	−0.156 *
		SO	19.3 ± 2.5 a	13.7 ± 2.5 c	15.2 ± 2.2 b	15.8 ± 2.0 b	14.8 ± 2.9 b,c	14.7 ± 2.5 b,c	˂0.001	−0.280 **
		ST	24.0 ± 2.6 a	19.3 ± 3.1 b	19.4 ± 2.2 b	19.5 ± 2.7 b	19.8 ± 1.7 b	19.3 ± 2.1 b	˂0.001	−0.315 **
	ΔE	SB	0.0 ± 0.0 a	9.3 ± 4.7 b	8.3 ± 5.0 b	8.1 ± 2.8 b	9.6 ± 4.3 b	9.4 ± 3.9 b	˂0.001	0.409 **
		SO	0.0 ± 0.0 c	10.8 ± 4.5 a	9.7 ± 5.8 b	7.7 ± 5.3 b	8.8 ± 4.1 b,c	10.4 ± 8.5 a,b	˂0.001	0.326 **
		ST	0.0 ± 0.0 b	9.6 ± 3.8 a	10.1 ± 4.9 a	9.1 ± 3.1 a	8.0 ± 3.9 a	8.6 ± 4.0 a	˂0.001	0.304 **

Different letters on the same row show statistical differences between means (*p* < 0.05). Pearson correlation (r) is significant at the 0.01 (** *p*) and 0.05 (* *p*) levels. n.s., not significant; SB, sardines in light brine; SO, sardines in vegetable oil; ST, sardines in tomato sauce.

**Table 4 foods-11-00991-t004:** Microbial quality of canned sardines in light brine, in vegetable oil and in tomato sauce during storage at 4 °C.

Quality Indicators	Microbiological Analysis (CFU/g)			
	Storage Time (Days)			
	0	1	2	7
Aerobic mesophiles	˂10	˂10	˂10	˂10
Aerobic psychrophiles	˂10^2^	˂10^2^	˂10^2^	* present but <4 × 10^2^
Aerobic thermophiles	˂10^2^	˂10^2^	˂10^2^	* present but <4 × 10^2^
Anaerobic mesophilic sulphite-reducing bacteria	˂10	˂10	˂10	˂10
Anaerobic thermophilic sulphite-reducing bacteria	˂10	˂10	˂10	˂10

* For sardines in light brine or vegetable oil. CFU, colony-forming unit.

## Data Availability

Data is contained within the article or Appendix A.

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
