# Peer review of "Safety and Quality of Canned Sardines after Opening: A Shelf-Stability Study"

_foods, 2022, doi:10.3390/foods11070991_

Round 1
Reviewer 1 Report
Manuscript ID: foods-1649118
Safety and quality of canned sardines after opening: a shelf-stability study
The authors present an article focused on the shelf life of canned sardines after opening. The topic is interesting because it provides to consumers useful information on the expiration date of opened foods. This information could be used as a potential tool to slowdown the food waste phenomenon; it is well known that the fish and seafood sector account about the 35 % of annual loss and waste (FAO, 2015) and home storage is identified as a crucial point along farm-to-fork chain.
In my opinion, the manuscript takes into account each point by studying all the phenomena which occur in the deterioration of food.
Line 10: Please delete “at”.
Line 11: Please replace “its” with “their”.
Line 11: Please delete “ultimately”.
Line 15: Please replace “its” with “their”.
Line 30: Please delete “moreover”.
Line 32: Please replace “micronutrients” with “macronutrients”.
Line 34-35: Please rewrite the sentence, it is not possible to understand the subject.
Line 39: Please replace “their nutritional features” with “fish nutritional features”.
Line 42: Please replace “while” with “and”.
Lines 45-46: Please replace “, being these effects widely documented in the literature” with “; these effects are widely documented in the literature”.
Line 47: Please introduce the canned products before to focus on their opening and the lack of consumption.
Line 50: Please replace “aims” with “aimed”.
Line 76: Please, not in Baltic the scientific name.
Line 81: Please replace “from” with “for”.
Line 86: Are the authors sure that the expression “drained sauce” is appropriate?
Line 90: Please delete the point.
Lines 91-93: Is it necessary to repeat this?
Line 95: Please replace “apart from” with “except for”.
Line 96: Please delete “on”.
Line 96: Please replace “allowed” with “left”.
Although the burden of microbiological contamination on shelf life has been amply demonstrated, the number of microbiological samplings is lower compared to those intended for the evaluation of physical and chemical parameters. Moreover, from the second to the seventh day of storage, no microbiological analyses were carried out. How did the authors think about evaluating the shelf life without to consider this period?
Line 110: Please replace “across” with “on”.
Line 117: Please replace “Fat estimation, fatty acids, and vitamin E” with “Estimation of fat, fatty acids, and vitamin E”.
Line 176: Please replace “based in” with “on the basis of”.
Lines 176-177: Please delete “Plates were daily checked for the presence of microorganism’s colonies”.
Lines 201-204: Please delete this sentence; it is redundant.
Lines 224-226: Please rewrite the sentence.
Lines 260: Please delete “the purpose of this study”.
I agree with the authors about the relevant differences among experimental groups in vitamin E content, however no statistical analysis has been performed to prove them.
Figure 2. Please check the font of the graph SO and adjust the graphical appearance of the storage time axis.
Figure 2 and Figure 3. Please invert the two figures.
Table 3. Please use italic for “p”.
Figure 3 (Called Sensorial analysis of canned sardines in light brine (SB), in vegetable oil (SO) and in tomato 403 sauce (ST) during storage at 4 °C, but referred to biogenic amines, ..). Please control the figure; there is a square behind the legend.
Figure 3. Please change the position; it should be copy in the paragraph “3.4. Sensorial performance”
Lines 416-419: The authors justified the presence of psychrophiles and thermophiles on the 7th day of storage in two types of product considering the handling during the sampling. Moreover, they hypostasized that the same condition could be contemplated in the consumers’ home after the opening of the canned fish and during the storage. However, we must consider the methodology of sampling adopted by the authors for microbiological analyses: “Before opening, all cans were disinfected with a solution containing iodine (4 %, m/v) 101 and ethanol (70 %, v/v) and manipulated under asepsis conditions, as recommended by 102 Landry et al. [9]. For each product (SB, SO and ST), four cans from the same batch were 103 combined in a sterile bag” (lines 101-104). According to this methodology, it is not possible to compare the two moments and the authors did not simulate the opening at home, because they did not consider the environmental microbiological contaminations that inevitably affect the shelf life of opened canned fish (as the authors specified in lines 432-434). Perhaps the authors should justify the choice of sampling the samples in this way to evaluate the shelf life of opened canned fish.
Lines 414 and 427: Please check the concentration value.
Line 453: Please replace “comprises” with “comprise”.
Line 454: Please replace “is” with “are”.
Lines 457-458: Please rewrite this sentence.
Line 460: Please replace “reduced” with “reduce the”.
Conclusion. The author should explain better the potential influence of their study on the food waste.

Reviewer 2 Report
Safety and quality of canned sardines after opening: a shelf-stability study
Ok
General evaluation
The manuscript is very interesting, but major revisions are needed.
Abstract
Line16 – I didn´t understand the sentence: “despite being considered nonedible up to the 7th day”. Explain better…
Introduction
Concise and well written.
M & M
Line – 82 to 83 – “Three of them were used for the physicochemical assays and further three for the sensorial analysis, over six sampling days (D0, D1, D2, D3, D4 83 and D7 = 3 × 2 × 6)”. 2 or 3 ????
Line 86 to 87 – “Drained sauce from one can of each type of fish preserve was also collected over six 86 sampling days (D0, D1, D2, D3, D4 and D7 = 1 × 6).” – For what ????
Line 94 - Plastic container suitable for food storage. – With air or without, with lip or without?
Better to join in the same item 2.2.2 and 2.4.
Results and Discussion
Line 224 to 226 - If there was no significant difference, you don't need to keep comparing treatments.
Line 224 to 226 - These differences can be related to the type of sauce (e.g., water, oil), the incorporation of vegetables (e.g., tomato), the compositional variation of fish (e.g., season, gender), and technological processing [25,26]. Idem.
Table 1 - I don't understand how you compared the treatment averages shown in Table 1. It's confusing.
Line 232 – 233 - In most fish species, post-mortem pH is commonly between 6.0 and 6.8 based on the initial glycogen content (affected by the nutritional status and fish activity before death) and accumulation of L-lactate [27]. - I believe that this discussion is not relevant to the present subject.
Is Figure 2 about biogenic amines or sensory analysis? There must be something wrong.
What is the difference between table 2 and 3?
The legend of figure 3 is wrong.
Line 439- sensorial quality decay was observed with storage likely due to dryness and progressing oxidation. – Dryness? But there was no change in moisture over time.
Conclusion
Line 482 to 483 – This is a discussion.
Line 489 to 498 - This is a discussion.
Round 2
Reviewer 1 Report
In my opinion, the manuscript can be accepted for publication.